# Characteristics of sickle cell patients with frequent emergency department visits and hospitalizations

Kyle Kidwell[1,2], Camila Albo[2], Michael Pope[2], Latanya Bowman[3], Hongyan Xu[4], Leigh Wells[3], Nadine Barrett[3], Niren Patel[3], Amy Allison[5], Abdullah Kutlar[3]*

1 Vanderbilt University Medical Center, Nashville, Tennessee, United States of America, 2 Medical College of Georgia at Augusta University, Augusta, Georgia, United States of America, 3 Sickle Cell Center, Augusta University, Augusta, Georgia, United States of America, 4 Department of Population Health Sciences, Augusta University, Augusta, Georgia, United States of America, 5 Georgia Cancer Center, Augusta University, Augusta, Georgia, United States of America

* akutlar@augusta.edu

**Data Availability Statement:** All relevant data are within the paper and its supporting files.

**Funding:** The authors received no specific funding for this work.

## Abstract

Vaso-occlusive episodes (VOEs) are a hallmark of sickle cell disease (SCD), and account for >90% of health care encounters for this patient population. The Cooperative Study of Sickle Cell Disease, a large study enrolling >3000 patients, showed that the majority of SCD patients (80%) experienced 0–3 major pain crises/year. Only a small minority (~5%) experienced ≥6 VOEs/year. Our study sought to further understand this difference in VOE frequency between SCD patients. We analyzed 25 patients (13M/12F, mean age of 28.8) with ≥6 ED visits or hospitalizations/year (high utilizers), and compared these with 9 patients (6M/3F, mean age of 37.6) who had ≤2 ED visits or hospitalizations/year (low utilizers). All subjects were given a demographic survey along with questionnaires for depression, anxiety, and Health Locus of Control. Each subject then underwent quantitative sensory testing (QST) with three different modalities: pressure pain sensitivity, heat and cold sensitivity, and Von Frey monofilament testing. Laboratory and clinical data were collected through subjects' medical records. CBC and chemistry analysis showed high utilizers had higher WBC (p<0.01), ANC (p<0.01), total bilirubin (p = 0.02), and lower MCV (p = 0.03). Opioid use (morphine equivalents) over the past 6 months was significantly higher in the high utilizer group (12125.7 mg vs 2423.1 mg, p = 0.005). QST results showed lower pressure pain threshold at the ulna (224.4 KPa vs 338.9 KPa, p = 0.04) in the high utilizer group. High utilizers also had higher anxiety (9.0 vs 4.6, p = 0.04) and depression scores (10.0 vs 6.0, p = 0.051). While the low utilizer group had higher education levels with more associate and bachelor degrees (p = 0.009), there was no difference in income or employment. These data show that many biological and psychosocial factors contribute to high health care utilization in SCD. A multi-disciplinary and multi-faceted approach will be required to address this complex problem.

**Competing interests:** The authors have declared that no competing interests exist.

## Introduction

Vaso-occlusive episodes (VOEs) are a hallmark of sickle cell disease (SCD), and account for over 90% of health care encounters for this patient population. Although an orphan disease (total number of patients in the US ~110,000), SCD has a huge medical-economic impact, with annual hospitalization costs of nearly $500 million by 2004 figures [1]. The Cooperative Study of Sickle Cell Disease (CSSCD) was a large natural history study, carried out in >23 Centers in the US, enrolling >3000 patients between 1977–93. An analysis of the frequency of VOE's in CSSCD showed that most patients (80%) experienced 0–3 major pain crises/year; this was similar across all genotypes of the disease (SS, SC, Sβ° thalassemia, Sβ⁺ thalassemia). Only a small minority (~5%) experienced ≥6 VOE's/year. Similar findings have been reported in subsequent smaller studies, with 5–10% of SCD patients having frequent ED visits and hospitalizations [2].

The factors leading to increased VOE's in certain SCD patients is complex and encompasses many interwoven components. A minority of SCD patients at the Augusta University Sickle Cell Center have frequent ED visits and hospitalizations, as defined by having ≥6 VOE's/year. Our study sought to further understand the difference in VOE frequency between these patients and SCD patients with less frequent VOE's. We performed a comprehensive analysis of demographic, clinical, laboratory, and psychosocial characteristics of 25 patients with ≥6 ED visits or hospitalizations/year (high utilizers), and compared these with 9 patients who had ≤2 ED visits or hospitalizations/year (low utilizers). Our goal was to highlight hematologic, biochemical, clinical and psychosocial differences between these two groups with the hope that specific interventions could be devised to reduce VOE frequency and improve the overall morbidity associated with this chronic disease.

## Materials/Methods

### Study design and subjects

This cross-sectional study was conducted between May 2016 and July 2016. Patients were enrolled during routine clinic visits at the Sickle Cell Center in Augusta, GA. Inclusion criteria for the study group (high utilizers) included: 1) Age ≥18 years old, 2) SCD (all genotypes), 3) ≥6 emergency department visits or hospitalizations for acute VOE's/sickle cell crisis pain within the last calendar year at the time of recruitment. Inclusion criteria for the control group (low utilizers) included: 1) Age ≥18 years old, 2) SCD (all genotypes), 3) ≤2 emergency department visits or hospitalizations for acute VOE's/sickle cell crisis pain within the last calendar year at the time of recruitment. Exclusion criteria for both groups included patients that were hospitalized within the last two weeks or patients that were transfused within the last 3 months prior to recruitment, as these could affect laboratory values and pain reporting during testing.

The Institutional Review Board of Augusta University approved this study. Written informed consent was obtained for each patient prior to testing, and a total of 34 patients were evaluated (25 in the study/high utilizer group and 9 in the control/low utilizer group).

### Data collection: Questionnaires

The enrolled subjects were first given a series of four questionnaires. The first two consisted of the PHQ-9 to screen for depression and the GAD-7 to screen for anxiety. Both the PHQ-9 and the GAD-7 are widely accepted as reliable and valid screening tools for depression and anxiety, respectively [3,4]. The third questionnaire was the Multidimensional Health Locus of Control, Form C, which is used to evaluate a psychosocial health component called the health locus of

control (HLOC). This questionnaire involves 18 statements, which patients rate on a scale from 1–6 representing strongly disagree (1), moderately disagree (2), slightly disagree (3), slightly agree (4), moderately agree (5), and strongly agree (6). These numbers are then added together to give an objective score in four categories: internal, chance, doctors, and other people. The higher the score in a category, the more a patient believes that component controls their disease. The fourth and final questionnaire consisted of a one page document created by the investigators of this study to evaluate each subject's education level, employment, household income, marital status, number of children, and number of people living in their primary residence.

### Data collection/Medical chart review

Blood samples for hematologic or biochemical analysis were not taken directly as part of this study, but often were collected during a patient's clinic visit on the date of study enrollment. If labs not drawn on the date of study enrollment, the most recent labs were used from the patient's prior clinic visit, which was usually 2 months before enrollment. Patients were well at the time of their clinic visit, so the lab work obtained was considered to be their steady state. We reviewed the following laboratory data: hemoglobin, hematocrit, white blood cell count, absolute neutrophil count, MCV, MCH, MCHC, platelet count, reticulocyte count, sodium, potassium, chloride, calcium, BUN, creatinine, total protein, albumin, AST, ALT, total bilirubin, LDH, CRP, ferritin, and hemoglobin electrophoresis including hemoglobin F levels. Hydroxyurea usage was documented for each patient with the corresponding dose prescribed. Duration of therapy or compliance with hydroxyurea was not measured as part of this study. Total opioid use over the last 6 months prior to recruitment was also determined via chart review, and the total of each medication was converted to milligrams of morphine equivalents to allow for comparison during analysis. Lastly, ED visits and hospitalizations over the last year prior to recruitment were calculated from the Augusta University Medical Center only, as the large majority of healthcare encounters for this population occur at our institution.

### Data collection: Quantitative sensory testing

Quantitative sensory testing was performed on each subject including pressure pain thresholds, heat and cold sensitivity, and Von Frey monofilament mechanical pain testing, performed in that order. To limit inter-observer variability, all quantitative sensory testing was performed by one of two investigators working on the study who were well versed in each testing protocol. The investigators performing the quantitative sensory testing completed the same equipment training prior to starting, but it should be noted that direct measurement of variability between observers was not performed. Patients were seated comfortably in a quiet room for the duration of testing.

### Pressure pain threshold testing

Pressure pain thresholds were determined using a handheld computerized algometer (AlgoMed, Medoc, Israel). Testing was done at three sites on the left side of the body in the following consecutive order: masseter, trapezius, and ulna. To begin, pressure was applied with the algometer and increased linearly at a constant rate. Subjects were given a button and instructed to push it when the pressure sensation turned to pain, with the computer recording the pressure pain threshold. There was no strict inter-stimulus interval between trials. A total of four trials at each site were performed and then averaged for analysis.

### Heat/Cold sensitivity testing

Heat and cold sensitivities were determined using a Q-sense computer-driven thermode attached to the left lower ventral forearm (Q-sense, Medoc, Israel). The baseline temperature of the thermode was 32˚C. Subjects were directed to push a button when they first felt the temperature change, emphasizing that this was not a measure of pain thresholds but of ability to detect heat and cold stimuli. The thermode would begin increasing or decreasing by 1˚C/sec from baseline until the patient pushed the button indicating that they detected a temperature change. After each trial, there was an inter-stimulus interval of 5 seconds before the next trial began. A total of four trials of both the heat and cold stimulus were performed and then averaged for analysis.

### Mechanical pain testing

Mechanical pain testing was done using a 300g Von Frey monofilament that is made to exert 300 grams of pressure upon bending. Three trials were performed for each subject on the dorsum of the right hand. The first trial consisted of a baseline pain measurement, where the monofilament was applied to the skin one time (until the filament bent to ensure proper pressure exertion), followed by a report of the level of pain perceived on a scale of 0–10. Two more trials were conducted after the baseline measurement at the same location. For each of the two trials, the monofilament was applied to the skin ten times at a rate of 1 application/second. The examiner listened to a metronome through a set of headphones to maintain consistent timing of filament placement. Following each of these trials, subjects were again asked to rate the pain they perceived on a scale of 0–10.

### Statistical methods

Two-sample t-tests were used to compare the mean values of continuous variables between the high utilizers and the low utilizers. Fisher's exact tests were used to compare the frequencies of categorical variables between the high utilizers and the low utilizers. All tests were two-sided and performed with R4.0.2 at 0.05 significance level.

## Results

### Group characteristics

A total of 34 patients were evaluated, with 25 in the study (high utilizer) group and 9 in the control (low utilizer) group. The average age of the subjects in the high utilizer group was 28.8 years old with an average of 15.6 ED visits/hospitalizations within the year prior to study enrollment. For the low utilizer group, the average age was 37.6 with an average of 0.44 ED visits/hospitalizations within the year prior to study enrollment. Table 1 provides a summary of the relevant demographics of the two study groups. Genotypes of both groups are shown in Table 2. All patients in the low utilizer group were Hb SS. Of the patients in the high utilizer

**Table 1. Demographic data for the two study groups.**

| Demographics | High Utilizer Group | Low Utilizer Group |
|---|:---:|:---:|
| Total subjects enrolled (n) | 25 | 9 |
| Male—n (%) | 13 (52) | 5 (56) |
| Female—n (%) | 12 (48) | 4 (44) |
| Average Age (Years) | 28.8 | 37.6 |
| Average hospitalizations/ED visits | 15.6 | 0.4 |

**Table 2. Represented sickle cell disease genotypes.**

| Genotype | High Utilizer Group | Low Utilizer Group |
|---|---|---|
| Hb SS | 19 | 9 |
| Hb SC | 4 | 0 |
| Hb SD-Los Angeles | 1 | 0 |
| Hb Sδβ-Thalassemia | 1 | 0 |

group, a large majority were Hb SS and Hb SC, with two other rare genotypes as noted in Table 2.

## Psychosocial components

Comparison of education level between the two groups revealed the low utilizer group had significantly more Associate's and Bachelor's degrees compared to the high utilizer group (p = 0.009). There was no significant difference between the two groups regarding income, employment, number of children, or number of people living in the household.

The mean anxiety score was 9.0 in the high utilizer group and 4.6 in the low utilizer group, with the difference being statistically significant (p = 0.039). The suggested cutoff for a positive screen with the GAD-7 is 8 from a large multi-center primary care study [3]. Of the 25 subjects in the high utilizer group, 13 (52%) screened positive with a score of 8 or greater. In contrast, only 2 subjects (22%) in the low utilizer group screened positive.

The difference in mean depression score was marginally statistically significant, with the high utilizer group averaging 10.0 and the low utilizer group averaging 6.0 (p = 0.051). A large study of 6,000 patients revealed that a cutoff score of 10 on the PHQ-9 yielded an 88% sensitivity and specificity for major depression [4]. In our study, 10 of the 25 subjects (40%) in the high utilizer group and 1 out of 9 subjects in the low utilizer group (11%) had a PHQ-9 score greater than 10.

The last psychosocial component we measured was the health locus of control (HLOC), which is displayed in Table 3. There was no significant difference in the HLOC sub-scores for internal, doctors, or other people. However, the chance HLOC sub-score was significantly higher in the high utilizer group compared to the low utilizer group.

## Laboratory components

A selection of the laboratory data obtained for each group is shown in Table 4. The average values for sodium, potassium, chloride, calcium, BUN, creatinine, total protein, albumin, AST, and ALT were not included as the differences between the two groups were not statistically significant. Only 2 out of 9 subjects in the low utilizer group had labs drawn for LDH and only 1 out of 9 had labs drawn for CRP, so these two labs could not be analyzed. Similarly, there was

**Table 3. Comparison of health locus of control (HLOC) sub-scores among SCD patients with high and low healthcare utilization.**

| HLOC Sub-score | High Utilizer Group (Average) | Low Utilizer Group (Average) | P-value |
|---|---|---|---|
| Internal (6–36) | 23.8 | 21.8 | 0.45 |
| Chance (6–36) | 19 | 13.6 | 0.047 |
| Doctors (3–18) | 13.7 | 12.9 | 0.61 |
| Other People (3–18) | 10.7 | 9 | 0.37 |

**Table 4. Comparison of laboratory data among SCD patients with high and low healthcare utilization.**

| Lab Component | High Utilizer Group (Average) | Low Utilizer Group (Average) | P-value |
|---|---|---|---|
| Hb (g/dL) | 9.3 | 9.57 | 0.66 |
| Hct (%) | 27.67 | 28.12 | 0.8 |
| WBC (thous/mm3) | 13.16 | 8.62 | <0.01 |
| ANC (thous/mm3) | 7.87 | 3.9 | <0.01 |
| MCV (fL) | 91.38 | 107.71 | 0.03 |
| MCH (pg) | 30.96 | 36.7 | 0.02 |
| MCHC (g/dL) | 33.8 | 34.07 | 0.42 |
| Retic (%) | 8.1 | 7.32 | 0.63 |
| Platelets (thous/mm3) | 364.48 | 322.11 | 0.23 |
| Total Bilirubin (mg/dL) | 4.23 | 2.27 | 0.02 |
| Fetal Hb (%) | 11.18 | 17.53 | 0.14 |

insufficient data for ferritin as only 5 out of 25 in subjects in the high utilizer group and 4 out of 9 subjects in the low utilizer group had this lab drawn.

Subjects in the high utilizer group had significantly higher white blood cell (WBC) counts, absolute neutrophil counts, and total bilirubin levels. lower mean corpuscular volumes (MCV) compared to the low utilizer group (Table 4 and Fig 1). The average MCV of the low utilizer group was 107.7, which is higher than normal assuming a reference range of 80–100 fL. It is well known that regular use of hydroxyurea (HU) increases MCV in patients with sickle cell disease. 77.8% (7/9) of subjects in the low utilizer group were prescribed HU at the time of enrollment, and they all had an MCV >100 fL. 76% (19/25) of subjects in the high utilizer group were prescribed HU at the time of study enrollment, which is similar in percentage to that of the low utilizer group. However, only 5/19 patients prescribed hydroxyurea in the high utilizer group had an MCV >100 fL.

## Quantitative sensory testing components

The results of pressure pain threshold testing are shown in Fig 2. Pain thresholds were lower at all three sites for the high utilizer group, but only the results at the ulna reached statistical significance (p = 0.042).

Mechanical pain testing with Von Frey monofilament revealed no significant difference between the two groups, as shown in Table 5. Of the 9 participants in the low utilizer group, 6 reported a baseline pain score of 0, with the remaining 3 reporting a pain score of 1. No patients in the low utilizer group reported a baseline pain score greater than 1. Of the 25 patients in the high utilizer group, 6 reported baseline pain scores greater than 1. Although this is a small sample size, these results agree with the pressure pain threshold testing, and demonstrate that sickle cell patients with frequent ED visits and hospitalizations likely have increased sensitivities to multiple pain modalities.

Heat and cold sensitivity testing revealed no significant difference between the two groups. The average cold temperature detection threshold between the high utilizer and low utilizer groups was 29.42°C and 28.63°C, respectively (p = 0.14). The average heat temperature detection threshold in the high utilizer and low utilizer groups was 36.42°C and 35.81°C, respectively (p = 0.25).

Lastly, the average total opioid use over the 6 months prior to recruitment in the high utilizer group was 12,125.6 mg morphine equivalents, while the average for the low utilizer group was 2423.1 mg morphine equivalents (p = 0.005).

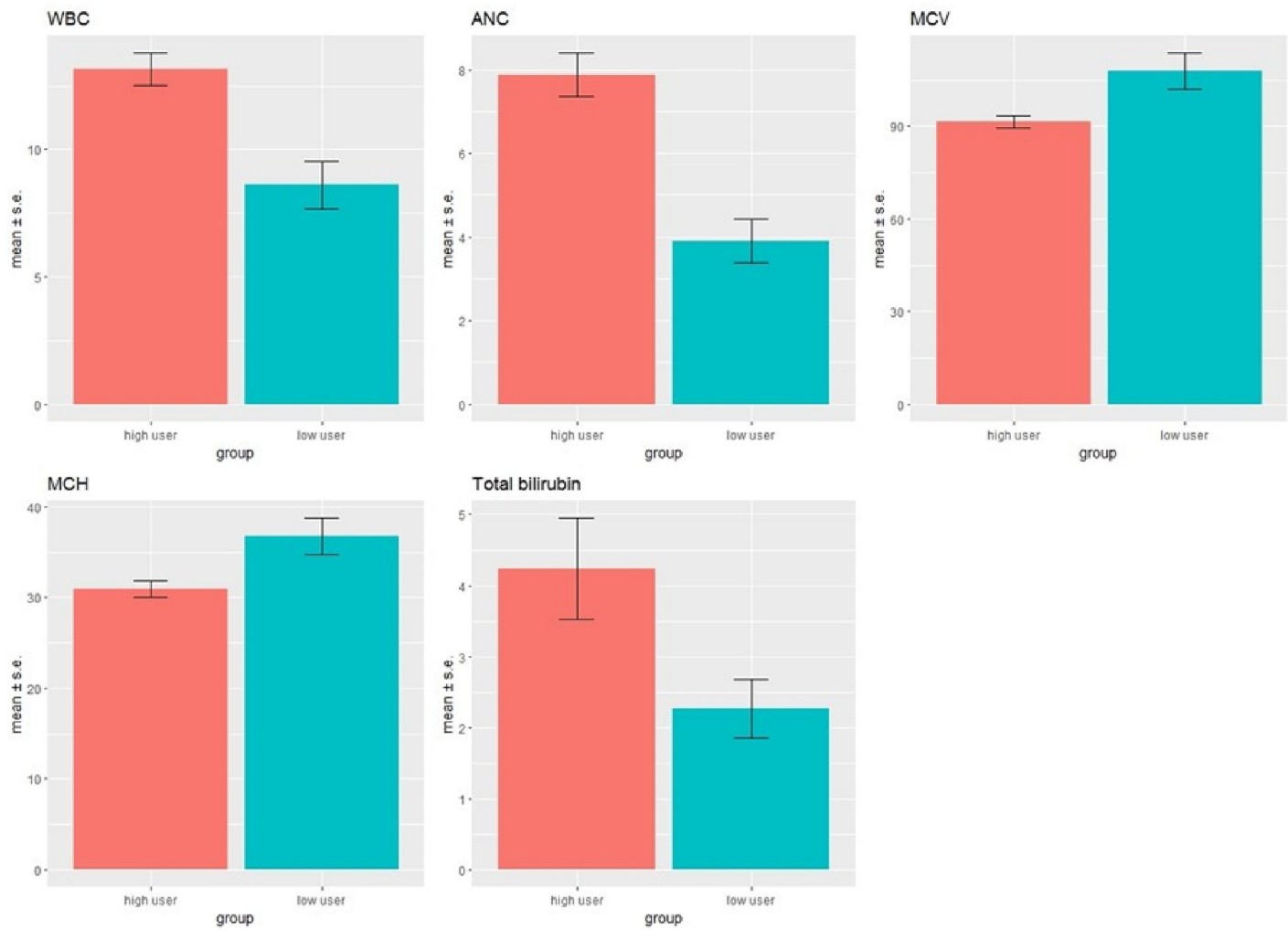

**Fig 1. Laboratory data in high utilizer (n = 25) vs low utilizer (n = 9) groups.** CBC and chemistry analysis showed high utilizers had higher WBC (p<0.01), ANC (p<0.01), total bilirubin (p = 0.02) and lower MCV (p = 0.03), MCH (p = 0.02) compared to the low utilizers.

## Discussion

Overall, we highlighted a few significant differences between SCD patients with frequent VOE's compared to those without frequent VOE's. First, we noted lower education levels with higher depression, anxiety, and HLOC chance sub-scores in the high utilizer group. We also found that patients with frequent VOE's had higher WBC counts, absolute neutrophil counts, and total bilirubin levels, suggesting more severe disease. Subjects in the high utilizer group had lower MCV values despite a similar rate of hydroxyurea prescription compared to the low utilizer group. Lastly, we demonstrated increased sensitivity to pressure pain at the ulna with concomitant higher opioid use in subjects from the high utilizer group.

Many prior studies have shown that depression is prevalent in SCD [5–7]. Health care costs for SCD patients with depression were found to be more than double those of SCD patients without depression, pinpointing depression as a major target for decreasing healthcare costs in SCD [7]. The same study notes that depression in SCD is associated with worse quality of life outcomes. Our study showed that the PHQ-9 depression screening tool scores were

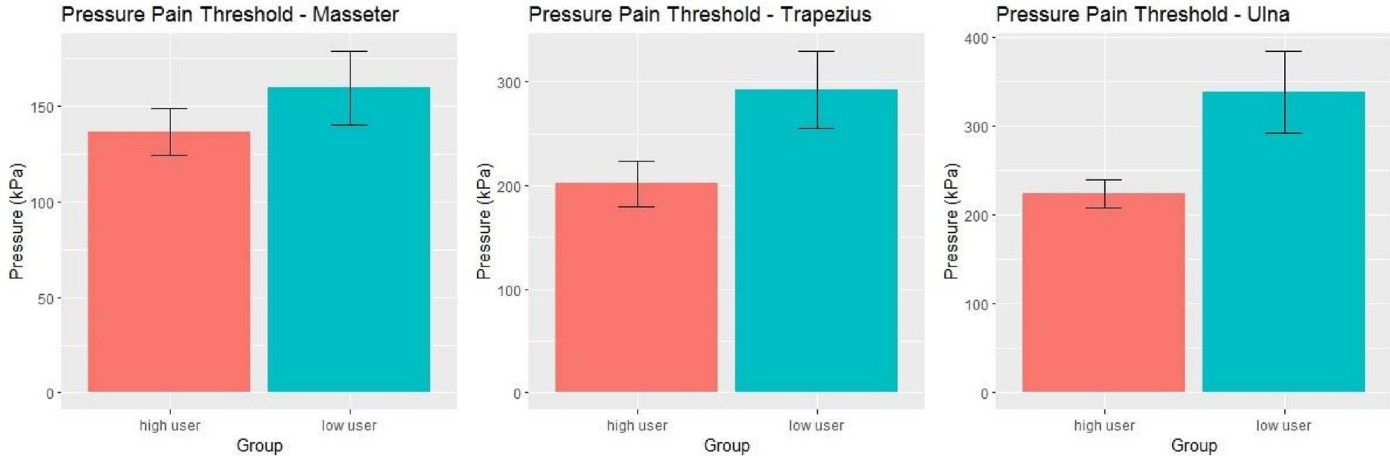

**Fig 2. Pressure pain threshold testing results in the high utilizer (n = 25) vs low utilizer (n = 9) groups.** Pressure was increasingly applied at a constant rate with a hand-held digital algometer 4 times at each site (masseter (A), trapezius (B), and ulna (C)). Subjects pushed a button when the pressure became painful, and the recordings were averaged. Average pressure pain thresholds were lower in the high utilizer group at all 3 sites, with a statistically significant difference at the ulna (p = 0.042).

significantly higher in persons with sickle cell disease patients who had frequent ED visits and hospitalizations. This suggests SCD patients who have frequent ED visits and hospitalizations may have higher rates of co-existing depression that could be contributing to poor control of their sickle cell disease. Due to the compounding evidence that depression correlates with low quality of life and higher health care utilization in SCD, we propose that clinicians should strongly consider screening for depression in all SCD patients.

The other two major psychosocial aspects we investigated were anxiety and the multidimensional health locus of control (HLOC). We found significantly higher scores on the GAD-7 anxiety screening tool in SCD patients with frequent health care utilization. The PiSCES project, which evaluated 308 adults with SCD, found that the prevalence of any anxiety disorder in these patients was 6.5%, which was much lower than the prevalence of depression [8]. In addition, the GAD-7 has a much higher negative predictive value than positive predictive value, with many patients who screen positive not being diagnosed with generalized anxiety disorder. While the evidence for anxiety compared to depression in SCD is not as strong, further work should be done to advance our understanding of the relationship between anxiety and SCD outcomes. In relation to HLOC scores, we found that patients in the high utilizer group had higher HLOC chance sub-scores. Interestingly, a study of the caregivers of children with SCD found an inverse relationship between adherence to treatment and chance HLOC sub-scores [9]. This indicates that the higher HLOC chance sub-scores found in patients with frequent VOE's and hospitalizations could negatively impact their adherence to proper disease management.

Leukocytes play a critical role in the pathogenesis of SCD related vaso-occlusion through interactions with both the vascular endothelium and red blood cells. Studies of sickle cell mice

**Table 5. Comparison of Von Frey Monofilament pain scores among SCD patients with high and low healthcare utilization.**

| Von Frey Monofilament (Pain Score 0–10) | High Utilizer Group (Average) | Low Utilizer Group (Average) | P-value |
|---|---|---|---|
| Baseline | 0.92 | 0.33 | 0.056 |
| Trial 1 | 1.92 | 1.33 | 0.39 |
| Trial 2 | 2.2 | 1.67 | 0.47 |

have shown that inhibition of leukocyte adhesion to the vascular endothelium can protect against vaso-occlusion [10,11]. From this, it can be inferred that higher leukocyte counts would increase the total WBC/endothelial cell interactions and lead to more frequent episodes of vaso-occlusion. Human studies have also highlighted the importance of leukocyte counts on clinical outcomes in SCD. Anyaegbu *et. al*, demonstrated that the clinical severity of SCD is associated with higher WBC counts and absolute neutrophils counts [12]. Furthermore, higher leukocyte counts are correlated with increased mortality, episodes of acute chest syndrome, incidence of hemorrhagic strokes, and even increased risk of frequent ED visits [13–16]. Our study findings of higher total WBC count and absolute neutrophil count in the high utilizer group are consistent with this prior data. This emphasizes the importance of leukocyte and absolute neutrophil counts not only as a marker of increased clinical severity in SCD, but also as a predictor of patients at risk for increased frequency of VOE's.

Hydroxyurea (HU), the first disease modifying agent approved for adults with SCD, results in up to 40% reduction in overall mortality from the disease [17]. In addition to reducing overall mortality, HU therapy can decrease the frequency of VOE's and hospitalizations [18]. Despite this, HU remains underutilized in SCD, with medication compliance posing a common problem. In a recent study at our center, we found that 26.3% of the 137 patients studied were non-adherent, based upon the lack of anticipated change in several laboratory parameters [19]. Our current study resulted in similar findings, as a comparable percentage of patients in both groups (76% of high utilizers and 77.8% of low utilizers) were prescribed HU. However, the two groups had differences in several lab parameters, including lower MCV and higher leukocyte counts in the high utilizer group. We also found lower levels of Hb F in the high utilizer group, but this failed to reach statistical significance. It should be noted again that the duration of hydroxyurea therapy and dosages were not directly analyzed as a part of this study, so this data should be interpreted with caution. Nonetheless, it suggests that compliance to hydroxyurea may be poor in SCD patients with frequent VOE's and hospitalizations. This underscores the need for solid patient education programs to help patients understand the importance and benefits of HU in controlling their disease.

Quantitative sensory testing (QST) has emerged as a useful method to determine differences in pain processing between individuals. Prior studies of QST in SCD have shown that these patients exhibit increased sensitivity to thermal pain thresholds and pressure pain thresholds compared to healthy controls [20,21]. While we did not measure thermal pain thresholds, thermal detection thresholds did not differ significantly between the two groups in our study. However, we did find that SCD patients with an increased frequency of VOE's had lower pressure pain thresholds at the ulna. The other two sites (masseter and trapezius) revealed similar lower pain thresholds but these failed to reach statistical significance. This is consistent with data from another QST study, which showed that measurements at the ulna achieved the best discrimination between sickle cell and control subjects [22]. However, while the prior studies compared SCD patients to healthy controls, our study revealed that there are significant differences in pressure pain thresholds among SCD patients. One theory to explain this difference in pain perception is the concept of opioid induced hyperalgesia [23]. Interestingly, the high utilizer group in our study had significantly higher opioid use over the past 6 months compared to the low utilizer group. Higher opioid use, combined with increased baseline pain sensitivity on pressure pain threshold testing, suggests that opioid induced hyperalgesia may predispose certain patients to develop more frequent acute painful crises, leading to an increased number of ED visits and hospitalizations. However, it is also possible that genetic or environmental factors contribute to a more severe phenotype, resulting in increased hospitalizations and total opioid use. Further studies should be done to evaluate the complex

relationship between opioid use, pain sensitivity, and VOE frequency and development of chronic pain in SCD.

One of the limitations of this study is that it was performed at a single center with a patient population mostly from Augusta, GA and the local surrounding area. The study would be more generalizable to the SCD population if the study was conducted with patients from multiple centers across the U.S. Additionally, our study period was limited by time constraints, which resulted in a relatively small sample size of 34 participants. This reduced the power of our study and hindered our ability to adequately detect differences between the two groups in some of the study variables. One further weakness of the study was that investigators were not blinded to which group the subjects were in during testing or analysis. Although this likely did not make a major difference in the study outcomes, it is a potential source of bias that we would like to point out for future studies that may build on this research.

Our study shows that multiple biologic and psychosocial factors contribute to high VOE frequency and health care utilization in SCD. A multi-disciplinary and multi-faceted approach, including a sound patient education and transition program, a concerted effort to increase adherence to HU therapy, and appropriate treatment of psychiatric comorbidities such as anxiety and depression will be required to address this complex problem.

## Supporting information

**S1 File.**
(DOCX)

**S1 Data.**
(XLSX)

## Acknowledgments

We would like to acknowledge the patients and staff at the Augusta University Sickle Cell Center who helped make this effort possible.

## Author Contributions

**Conceptualization:** Abdullah Kutlar.

**Data curation:** Kyle Kidwell, Camila Albo, Michael Pope, Nadine Barrett, Niren Patel.

**Formal analysis:** Hongyan Xu, Niren Patel, Amy Allison.

**Investigation:** Camila Albo, Michael Pope, Latanya Bowman, Abdullah Kutlar.

**Methodology:** Amy Allison.

**Project administration:** Leigh Wells, Abdullah Kutlar.

**Resources:** Latanya Bowman, Leigh Wells, Nadine Barrett, Amy Allison.

**Supervision:** Latanya Bowman, Leigh Wells.

**Validation:** Hongyan Xu.

**Writing – original draft:** Kyle Kidwell.

**Writing – review & editing:** Hongyan Xu, Abdullah Kutlar.

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
