## [Decision Letter · Decision Letter 0]

27 Oct 2020

PONE-D-20-26173

Characteristics of Sickle Cell Patients with frequent Emergency Department visits and Hospitalizations

PLOS ONE

Dear Dr. Kutlar,

Thank you for submitting your manuscript to PLOS ONE. After careful consideration, we feel that it has merit but does not fully meet PLOS ONE’s publication criteria as it currently stands. Therefore, we invite you to submit a revised version of the manuscript that addresses the points raised during the review process.

Of special note is the issue of small sample size. How was this sample collected? Were any sample size calculations done and is the sample adequate to make any determinations confidently.

We look forward to receiving your revised manuscript. 

Kind regards,

Monika R. Asnani, DM, PhD

Academic Editor

PLOS ONE

Journal Requirements:

2. Please provide additional details regarding participant consent. In the ethics statement in the Methods and online submission information, please ensure that you have specified both (1) whether consent was informed and (2) what type you obtained (for instance, written or verbal, and if verbal, how it was documented and witnessed).

3. Please include additional information regarding the survey or questionnaire that was created by the authors and used in the study and ensure that you have provided sufficient details that others could replicate the analyses. For instance, if it is not under a copyright more restrictive than CC-BY, please include a copy, in both the original language and English, as Supporting Information.

4. Please list the name and version of any software package used for statistical analysis, alongside any relevant references.

5. In your Results section, please include a table of relevant participant demographic details.

Reviewers' comments:

Reviewer's Responses to Questions

**Comments to the Author**

1. Is the manuscript technically sound, and do the data support the conclusions?

Reviewer #1: Yes

Reviewer #2: Yes

Reviewer #3: Partly

2. Has the statistical analysis been performed appropriately and rigorously? 

Reviewer #1: Yes

Reviewer #2: Yes

Reviewer #3: No

3. Have the authors made all data underlying the findings in their manuscript fully available?

Reviewer #1: Yes

Reviewer #2: Yes

Reviewer #3: No

4. Is the manuscript presented in an intelligible fashion and written in standard English?

Reviewer #1: Yes

Reviewer #2: Yes

Reviewer #3: Yes

5. Review Comments to the Author

Reviewer #1: In this manuscript, Abdullah Kutlar and colleagues analyzed markers of vaso-occlusive episodes and frequent hospitalization among patients with sickle cell disease (SCD). The authors compared SCD patients with more than 6 emergency department (ED) visits per year (high utilizers) to the SCD patients who had less than two ED per year (low utilizers). They found that high utilizers had higher WBC, ANC and bilirubin and lower MCV. Also, high utilizers had lower pain threshold measured by Von Frey monofilament testing, consumed more opioids and had higher anxiety and depression scores. Overall, this is an informative study that may help to identify high utilizers that can be hospitalized without going through the expensive ER procedure. My only concern is that the data in Table 3 that show significant difference (i.e. WBC, ANC, MCV and MCH) would be better shown in the form of graphs.

Major:

1. Please show a graph with plotted WBC, ANC, MCV and MCH values to corroborate the data in Table 3.

Minor

1. Page 7, lane 161: “Two-sample t-tests were used to compared “ should be “Two-sample t-tests were used to compare “

Reviewer #2: Thank you for the opportunity to review this important paper. This study sought to understand the differences between patients with sickle cell disease considered as high or low utilizers of medical care as defined by the frequency of vaso-occlusive crises for which medical care was sought. Vaso-occlusive crisis is a very common event associated with sickle cell disease and therefore the study is well justified. The paper is well written and shows the extensive work that was put into this very relevant study.

However revisions are needed:

The Title: I suggest a revision to " Characteristics of persons with sickle cell disease with frequent Emergency Department ....

The phrase "Sickle Cell Patients " does not give an indication that a variety of genotypes were studied and should not be used.

see also line 200 - patients with sickle cell disease.

Reference#1: I would suggest that the primary reference, Steiner et al be used. The data in the reference ( Singh et al could be used separately to support the text.

The aims could be stated more specifically: For example what differences were being examined were they genotype, hydroxyurea use, previous opioid use, haematological and biochemical variables etc.

Typographical error - line 73 - "Hopes" please change to "Hope";

Methods: line 87 - change to "within the last calendar year at the time of recruitment"

Line89 - change to within the last 3 months prior to recruitment.

Line 110: Change to " Blood Samples for Haematological or biochemical analysis was not taken as a part of the study" or a similar phrase

Please indicate the period of time that results were accepted retrospectively from the time of enrollment. Please state whether patients were well at the time these results were documented and if the results obtained were considered to be "steady state results"

The examiners conducting quantitative sensory testing were well experienced and the results were generated from only 2 examiners however please state whether tests were done for inter-observer reliability.

Results: Please state the average duration of hydroxyurea therapy in patients on hydroxyurea and also if they were at maximum tolerated dose.

line 200- 202 should be placed in the discussion

Line 252-252-256 please move this to the discussion

lines 265 -267 - please move tis to the discussion

The legend submitted for figure 1 can be more concise. Please consider making this shorter.

Was a multivariate analysis of the main outcome considered?

Was the use of hydroxyurea between groups statistically significant? Please state this.

Discussion: The findings regarding hydroxyurea should be made cautiously. We are not told how long patients were on HU or if they were at maximum tolerated dose to assess parameters such as MCV.

The observations regarding hydroxyurea use even though they may be interesting should be stated a bit more cautiously as details concerning use was not documented in the paper.

Generally, the discussion needs to be more concise and directed on findings from the study.

Reviewer #3: Characteristics of Sickle Cell Patients with frequent Emergency Department visits and

Hospitalizations

Summary

The authors report on a small single site study in which they compared patients who had frequent A&E visits and admissions >6 visits per year to those who had <2. They examined associations with psychosocial indices with questionnaires and performed quantitative sensory testing. Chart reviews were used to ascertain laboratory and clinical data.

They identified group differences in psychosocial indices, pain perception as well as labora itory and clinical indices and posited an explanation for the associations reported.

Minor comments:

Introduction: Lin 58- word missing. Should be “carried out in”.

Figure:

- It should be edited to improve contrast and crispness of the text.

- The statistically significant finding should be indicated on the figure.

Other comments:

Sample size

Researchers have not given any reasons for including 25 high utilization and 9 low utilization subjects; on what basis was number of cases and controls decided? Was this an audit of all eligible patients? How many refused? Was there a primary outcome used to calculate a required sample size? What determined the length of time for subject accrual?

Methods:

What was the range of the period of time between blood investigations and the study visit?

Did the subjects access all their care at the study sites? Could they have used prescription opioids not accounted for in the chart review?

Were there any other methods which could have been used to assess sensation? Electronic von Frey has been reported to be more reliable and rapid than VFM in exploring mechanical pain thresholds”

Statistical methods:

- Was any adjustment made for multiple comparisons?

- T tests were done. Were the distributions of the normal variables normal?

- It would be helpful if the categorization of variables was provided. For example, how was education categorized?

- Was the absence of multivariate analysis a function of the sample size?

Results:

Some aspects of the results are interpretations of the data and may be better placed in the discussion. For example “The combination of lower pain thresholds and considerably higher opioid use in the high utilizer group suggests that opioid induced hyperalgesia may be contributing to increased pain sensitivity in these patients. Increased baseline sensitivity to pain could predispose SCD patients to developing more frequent acute painful crises, leading to the increased number of ED visits and hospitalizations seen in patients from the high utilizer group.”

Discussion:

On what basis was an assessment made of the direction of relationships? Cross-sectional studies demonstrate associations but the direction of relationships is usually elucidated by longitudinal assessment.

Authors have suggested that perhaps “opioid induced hyperalgesia may be contributing to increased pain sensitivity in these patients. Increased baseline sensitivity to pain could predispose SCD patients to developing more 66 frequent acute painful crises, leading to the increased number of ED visits and hospitalizations seen in patients from the high utilizer group”. Could it be that genetic or environmental factors cause more severe disease, with unpredictable and more frequent, requiring more therapy with opioids, missing of school and work and diminished vocational outcomes? This pattern could then lead to anxiety, depression and feelings of being at the mercy of chance; along with indicators if heightened inflammation and hemolysis. The occurrence currently of altered pain sensitivity may not be a long standing baseline but may have changed over time. The authors should consider other possible explanations for their findings.

The authors should discuss the reasons why only the ulnar measurement showed statistically significant differences. Could this have been related to sample size?

The authors highlight poor adherence to hydroxyurea, particularly in patients with severe disease and psychosocial stress. They posit that additional knowledge is needed to change behavior. Though not central to their paper, they may also suggest other interventions bit solely based on enhancing knowledge.

6. PLOS authors have the option to publish the peer review history of their article (what does this mean?). If published, this will include your full peer review and any attached files.

Reviewer #1: No

Reviewer #2: **Yes: **Angela E Rankine-Mullings

Reviewer #3: No

---

## [Author Response · Author response to Decision Letter 0]

17 Dec 2020

Reviewer 1:

1. WBC, MCV and ANC values have been plotted to corroborate the data in Table 3

2. Page 7, line 161: “… t-tests were used to compared” has been corrected to “…compare”

Reviewer 2:

1. We kept the term “Characteristics of Sickle Cell Patients…” in the title, because we used the term “sickle cell patients” broadly to encompass all genotypes (SS, SC etc) as indicated in the Table showing the demographic features 

2. Steiner et al has been used as reference 1

3. A sentence has been added to clearly specify the aims, on page 3, lines 72-73

4. Typographical error ‘hopes” has been corrected to “hope”, line 74

5. Line 87, changed to “…within the last calendar year at the time of recruitment”

6. Line 91 “…within the last 3 months prior to recruitment” added.

7. Line 115-117: “Blood samples for ……. as part of the study” added, and the period of time that results were accepted retrospectively has been clarified as the time of previous clinic visit, which is usually 2 months prior, and also it was confirmed that the subjects were at “steady state” at that time. 

8. A statement has been added to specify training of the examiners for quantitative sensory testing to address interobserver reliability (Lines 137-138)

9. Duration of HU therapy was not specified for the subjects enrolled; however, those who were prescribed HU were “on HU” for >1 year

10. Lines 200-202, 252-256 and 265-267 have been moved to discussion. Thanks for the suggestion.

11. Legend for Figure 1 has been revised and shortened.

12. Multivariate analysis of the main outcome was not considered due to limited sample size.

13. HU use was 76% in the frequent utilizer group, and 77.8% in the non-frequent utilizer group; the difference was not statistically significant (p=0.6367 with Chi-squared test). 

14. HU use is based upon prescription of the medication, and does not take into account issues related to adherence. The discussion with regards to HU has been modified along the lines suggested by the reviewer.

Reviewer 3:

1. Line 58: “..carried out in” added

2. The figure has been revised to improve the contrast and significant findings have been indicated.

3. The sample size was not based on a pre-calculated power analysis. The number of high utilizers and low utilizers were selected from patients who receive their care at our institution. Frequent utilizers were defined as those with >6 ED visits/hospitalizations in a year, based upon CSSCD data, as mentioned in the introduction. 

4. For the range of time between blood draw and study visit, see item #7 in response to Reviewer 2’s comments above

5. It is highly unlikely that the subjects received opioid prescriptions from other providers/institutions, as they were closely monitored in the Georgia Prescription Monitoring System.

6. We did not use electronic Von Frey assessment.

7. We did not perform adjustment for multiple comparisons due to limited sample size.

8. We checked for normality with Kolmogorov-Smirnov (K-S) test before the t-tests and K-S test did not reject the normality assumption.

9. The absence of multivariate analysis was a result of the sample size.

10. Some sections in the results have been moved to discussion, where they are more appropriate. See also item #10 in response to Reviewer 2

11. Discussion has been modified (Lines 424-429) to include the possible role of genetic and environmental factors contributing to disease severity and high utilization, as suggested by the reviewer.

12. The observation that only the ulnar measurement in QST showed significant difference between two groups could be a reflection of limited sample size.

---

## [Decision Letter · Decision Letter 1]

22 Jan 2021

PONE-D-20-26173R1

Characteristics of Sickle Cell Patients with frequent Emergency Department visits and Hospitalizations

PLOS ONE

Dear Dr. Kutlar,

Thank you for submitting your manuscript to PLOS ONE. After careful consideration, we feel that it has merit but does not fully meet PLOS ONE’s publication criteria as it currently stands. Therefore, we invite you to submit a revised version of the manuscript that addresses the points raised during the review process.

We look forward to receiving your revised manuscript.

Kind regards,

Monika R. Asnani, DM, PhD

Academic Editor

PLOS ONE

Reviewers' comments:

Reviewer's Responses to Questions

**Comments to the Author**

1. If the authors have adequately addressed your comments raised in a previous round of review and you feel that this manuscript is now acceptable for publication, you may indicate that here to bypass the “Comments to the Author” section, enter your conflict of interest statement in the “Confidential to Editor” section, and submit your "Accept" recommendation.

Reviewer #1: All comments have been addressed

Reviewer #2: All comments have been addressed

Reviewer #3: (No Response)

2. Is the manuscript technically sound, and do the data support the conclusions?

Reviewer #1: Yes

Reviewer #2: Yes

Reviewer #3: Yes

3. Has the statistical analysis been performed appropriately and rigorously? 

Reviewer #1: Yes

Reviewer #2: Yes

Reviewer #3: Yes

4. Have the authors made all data underlying the findings in their manuscript fully available?

Reviewer #1: Yes

Reviewer #2: Yes

Reviewer #3: Yes

5. Is the manuscript presented in an intelligible fashion and written in standard English?

Reviewer #1: Yes

Reviewer #2: Yes

Reviewer #3: Yes

6. Review Comments to the Author

Reviewer #1: (No Response)

Reviewer #2: The manuscript seeks to determine factors which contribute to high health care utilization in sickle cell disease which is a very important topic. The author's responses are satisfactory and the revisions adequate, however please see two minor comments:

1. Please correct typos in lines 42, 228 and 324.

2. Please clarify labs further in manuscript as follows: ....from the patient’s prior clinic visit ....Please add "which was usually 2 months before enrollment". The latter comment was in the response to the reviewer and is duly noted but should also be placed in the manuscript.

Reviewer #3: The sentence added does not obviate the benefit of inter observer reliability testing. This is a limitation which the researchers should acknowledge.

7. PLOS authors have the option to publish the peer review history of their article (what does this mean?). If published, this will include your full peer review and any attached files.

Reviewer #1: No

Reviewer #2: **Yes: **Angela E Rankine-Mullings

Reviewer #3: No

---

## [Author Response · Author response to Decision Letter 1]

2 Feb 2021

Reviewer 2:

The manuscript seeks to determine factors which contribute to high health care utilization in sickle cell disease which is a very important topic. The author's responses are satisfactory and the revisions adequate, however please see two minor comments:

1. Please correct typos in lines 42, 228 and 324.

Typos in lines 42, 228and 324 have been corrected. We thank the reviewer for noting this.

2. Please clarify labs further in manuscript as follows: ....from the patient’s prior clinic visit ....Please add "which was usually 2 months before enrollment". The latter comment was in the response to the reviewer and is duly noted but should also be placed in the manuscript.

The phrase “…which was usually 2 months before enrollment” was added in line 112

 Reviewer 3:

The sentence added does not obviate the benefit of inter observer reliability testing. This is a limitation which the researchers should acknowledge.

A sentence acknowledging the possibility of interobserver variability has been added to the discussion

---

## [Editor Report · Decision Letter 2]

5 Feb 2021

Characteristics of Sickle Cell Patients with frequent Emergency Department visits and Hospitalizations

PONE-D-20-26173R2

Dear Dr. Kutlar,

We’re pleased to inform you that your manuscript has been judged scientifically suitable for publication and will be formally accepted for publication once it meets all outstanding technical requirements.

Kind regards,

Monika R. Asnani, DM, PhD

Academic Editor

PLOS ONE
---

## [Editor Report · Acceptance letter]

9 Feb 2021

PONE-D-20-26173R2 

Characteristics of sickle cell patients with frequent emergency department visits and hospitalizations 

Dear Dr. Kutlar:

I'm pleased to inform you that your manuscript has been deemed suitable for publication in PLOS ONE. Congratulations! Your manuscript is now with our production department. 

Kind regards, 

on behalf of

Prof. Monika R. Asnani 

Academic Editor

PLOS ONE